# Differential Amperometric Microneedle Biosensor for Wearable Levodopa Monitoring of Parkinson’s Disease

**DOI:** 10.3390/bios12020102

**Published:** 2022-02-07

**Authors:** Lu Fang, Hangxu Ren, Xiyu Mao, Shanshan Zhang, Yu Cai, Shiyi Xu, Yi Zhang, Lihua Li, Xuesong Ye, Bo Liang

**Affiliations:** 1Department of Automation, Hangzhou Dianzi University, Hangzhou 310018, China; Fanglu@hdu.edu.cn (L.F.); 20061135@hdu.edu.cn (Y.Z.); 2Biosensor National Special Laboratory, Key Laboratory of Biomedical Engineering of Ministry of Education, College of Biomedical Engineering & Instrument Science, Zhejiang University, Hangzhou 310058, China; 21815097@zju.edu.cn (H.R.); 21915002@zju.edu.cn (X.M.); 12015019@zju.edu.cn (S.Z.); 3130104519@zju.edu.cn (Y.C.); xushiyi@zju.edu.cn (S.X.)

**Keywords:** Parkinson’s disease (PD), levodopa (L-Dopa), minimal-invasive, flexible differential microneedle array (FDMA), biosensor

## Abstract

Levodopa (L-Dopa) is considered to be one of the most effective therapies available for Parkinson’s disease (PD) treatment. The therapeutic window of L-Dopa is narrow due to its short half-life, and long-time L-Dopa treatment will cause some side effects such as dyskinesias, psychosis, and orthostatic hypotension. Therefore, it is of great significance to monitor the dynamic concentration of L-Dopa for PD patients with wearable biosensors to reduce the risk of complications. However, the high concentration of interferents in the body brings great challenges to the in vivo monitoring of L-Dopa. To address this issue, we proposed a minimal-invasive L-Dopa biosensor based on a flexible differential microneedle array (FDMA). One working electrode responded to L-Dopa and interfering substances, while the other working electrode only responded to electroactive interferences. The differential current response of these two electrodes was related to the concentration of L-Dopa by eliminating the common mode interference. The differential structure provided the sensor with excellent anti-interference performance and improved the sensor’s accuracy. This novel flexible microneedle sensor exhibited favorable analytical performance of a wide linear dynamic range (0–20 μM), high sensitivity (12.618 nA μM^−1^ cm^−2^) as well as long-term stability (two weeks). Ultimately, the L-Dopa sensor displayed a fast response to in vivo L-Dopa dynamically with considerable anti-interference ability. All these attractive performances indicated the feasibility of this FDMA for minimal invasive and continuous monitoring of L-Dopa dynamic concentration for Parkinson’s disease.

## 1. Introduction

Parkinson’s disease is a long-term heterogeneous neurodegenerative brain disorder [1,2]. Its main symptoms include tremor, slowness of movement, stiff muscles, and unsteady walk, which seriously affect the patient’s quality of life [3]. According to the Parkinson’s Foundation, there are more than 10 million people worldwide currently living with PD. A striking feature of PD is the loss of dopaminergic neurons in the substantia nigra pars compacta and leads to a reduction in dopamine levels in the striatum [4]. L-Dopa, which can cross the blood-brain barrier and be converted to dopamine [5], has been considered as one of the most effective therapies available for Parkinson treatment since the 1960s [6] and is acknowledged as the ‘gold standard’ of symptomatic efficacy in the drug treatment of PD [7,8]. While L-Dopa greatly improves the patient’s motor symptoms, a major problem of L-Dopa treatment is that the therapeutic window is narrow. The efficacy decreases as the disease progresses, and low doses of L-Dopa will lead to complications such as dyskinesias, psychosis, and orthostatic hypotension [9], which occur in up to 80% of patients [10,11]. Due to the differences in drug sensitivity and metabolic level of patients, doctors need to adjust the dosage of each patient individually to ensure its efficacy and reduce the side effects associated with dose fluctuation of L-Dopa. Therefore, it is of great significance to monitor the dynamic concentration of L-Dopa in patients during treatment.

Researchers have carried out significant work toward the detection of L-Dopa based on various principles, such as spectrophotometry, capillary electrophoresis, fluorescence spectroscopy, high-performance liquid chromatography, and electrochemistry [12,13,14,15,16]. Among them, the electrochemical method has received widespread attention and plays an important role in determination of biological and environmental analysis. Moreover, electrochemical sensors are particularly suitable for wearable and in vivo monitoring owning to their advantages of small size, high-sensitivity, and good linearity [17,18,19]. A large number of studies have reported the application of wearable or implantable electrochemical sensors for the real-time dynamic analysis of biomolecules such as glucose, neurotransmitters, and lactic acid, among others [20,21,22,23,24,25,26,27,28]. The continuous monitoring of L-Dopa toward Parkinson management with electrochemical sensors has also been reported recently [29,30,31,32]. The research group of Ali Javey presented a wearable sweatband that quantitatively tracked L-Dopa dynamics in the body [31]. The sensor reached an ultralow LOD of 1.25 μM in sweat, with a linear range of 0–20 μM, by using gold dendritic nanostructures to increase surface area, but it was a problem to obtain enough sweat on older PD patients for real-time detection. Meanwhile, a dual-mode microneedle sensing platform for continuous minimally invasive electrochemical detection of L-Dopa was reported by Joseph Wang [30]. It could record two different types of L-Dopa signal on two electrodes at the same time and offered a built-in redundancy. However, the performance of the sensor was only evaluated in artificial interstitial fluid with an ex vivo skin model, and there is still much work to do towards in vivo application.

Selectivity, or anti-interference ability, is an important property of biosensors, especially for electrochemically monitoring L-Dopa in vivo, because the in vivo concentration of L-Dopa is extremely low compared to other electroactive interferences (e.g., ascorbic acid, uric acid, acetaminophen) in physiological environments [33,34,35]. These interferences are easily oxidized on the electrode surface and result in inaccuracy of the sensor signal. Researchers have tried various methods to improve the selectivity of the sensor. For example, the use of the specific recognition between biomolecules (such as enzyme-substrate, antibody-antigen, ligand-receptor, and molecularly imprinted polymers-imprinted molecules) could improve the specificity of the sensor [36]. Furthermore, the use of a permselective membrane could eliminate electroactive interfering species and reduce signal interference [37,38], and a lower operating potential of the biosensor could effectively reduce the oxidation current of the interferent [39]. Although these commonly used anti-interference methods can improve the sensor’s selectivity, they are not enough for in vivo L-Dopa monitoring, so there still remains the challenge of developing an L-Dopa sensor with good selectivity and accuracy.

In this study, we reported a novel method based on differential structure to improve the selectivity and accuracy of the minimal-invasive microneedle array towards continuous monitoring of L-Dopa. As illustrated in Figure 1, the flexible differential microneedle array (FDMA) was fabricated on a flexible polyimide substrate with six electrodes, which can be divided into two groups: group 1 (working electrode, WE1, counter electrode, CE1, and reference electrode, RE1) and group 2 (working electrode, WE2, counter electrode, CE2, and reference electrode, RE2). The surface of WE1 was modified with tyrosinase, while the WE2 surface had no tyrosinase. Therefore, during in vivo detection, L-Dopa would be catalyzed by tyrosinase on the outer layer of WE1 and oxidized to dopaquinone. Therefore, WE1 only responded to electroactive interfering substances. However, because there was no tyrosinase on the surface of WE2, both L-Dopa and interferents could pass through the outer layers of WE2 and reach the Au nanodendrites catalyzed layer. Therefore, under the applied working potential, WE2 could respond to L-Dopa and interferents by directly electrochemically oxidizing L-Dopa and interferents. The differential current response of these two electrodes was related to the concentration of L-Dopa by eliminating the common mode interference. This differential approach effectively addressed the challenges for low-concentration substance detection under the influence of high-concentration interfering substances.

The FDMA showed excellent selectivity and good linear current response toward L-Dopa in PBS and bovine serum, together with a considerable stability. Furthermore, the FDMA exhibited strong toughness and favorable response stability upon penetration though rat skin. Finally, the sensor displayed a rapid response to in vivo concentration fluctuations of L-Dopa with good anti-interference ability, including interference from chemical substances and movement. All these attractive performances indicated the potential use of the FDMA as a minimal invasive biosensor for dynamically monitoring the drug concentration in patients with Parkinson’s disease. It could provide a basis for healthcare professionals to adjust the drug dosage, realize the personalized treatment of Parkinson’s disease, and improve patients’ quality of life.

## 2. Materials and Methods

The materials and reagents, instruments, and in vitro and in vivo evaluation methods of the biosensor are shown in the Appendix A.

### 2.1. Preparation of the Microneedle Electrode

A stainless steel (SS) microneedle (Φ = 160 μm, 7 mm in length) was chosen as the electrode substrate in this study. SS electrodes were ultrasonically polished and cleaned in 80 mg/mL aluminum oxide (Φ = 50 nm) and deionized (DI) water for 10 min, respectively. After that, an electrochemical cleaning process was carried out by cyclic voltammetry (−0.4 V to 0.6 V, 10 cycles, scan rate 50 mV/s) until there was no significant redox peak (Appendix A). Then, the SS electrodes were rinsed with DI water and dried at room temperature.

A differential structure was used in this study to reach the improved anti-interference ability of the sensor. As shown in Figure 1E and Appendix A, the working electrode WE1 and WE2 were modified with four layers: Au nanodendrite catalytic layer, Nafion layer, PANI/enzyme layer, and PU layer. The difference between WE1 and WE2 was that there was tyrosinase in the PANI/enzyme layer of WE1, but not in the PANI/enzyme layer of WE2. The electrode modification process is shown in Figure 2A: firstly, the cleaned SS electrode was immersed in 10 mM HAuCl_4_/0.1 M HCl solution and a constant voltage of 0 V (vs Ag/AgCl) was applied for 300 s for the electrodeposition of Au nanodendrites. After that, 1% Nafion solution diluted in ethanol was applied to the as-synthesized Au nanodendrites and dried in air. For a better immobilization of enzyme, a polyaniline (PANI) thin film was electropolymerized on the SS/Au/Nafion electrode surface by galvanostatic with 0.1 mA/cm^2^ for 100 s in 0.4 M aniline/1 M HCl solution. The electrode was then immersed in deionized water for 30 min and dried at room temperature. To immobilize the enzyme, the WE1 was immersed in a solution of 5 mg/mL tyrosinase and 40 mg/mL BSA, and the WE2 was immersed in a solution of 45 mg/mL BSA. Both electrodes were stored at 4 °C for 24 h. After adsorption, the electrodes were rinsed in deionized water. Next, both working electrodes were placed in a sealed bottle with 20 μL of 25% glutaraldehyde solution and placed in an incubator at 37 °C for 30 min cross-linking. The final modification step for the working electrode involved the coating of PU, which enhanced the long-term stability and antifouling features of the electrochemical sensor. Three percent (*w*/*w*) PU solution was prepared by dissolving PU in 2% dimethyl formamide (DMF) and 98% tetrahydrofuran (THF). A circular ring with a diameter of 2 mm was dipped into the PU solution to form a membrane of micron-grade thickness in the middle of the ring. The electrode was then passed through the ring, and a PU membrane was formed uniformly on the electrode surface. After being dried in air for 2 h, both electrodes were stored in 0.1 M PBS at 4 °C and were ready for use.

### 2.2. Preparation of the Flexible Differential Microneedle Array

An assembled FDMA was fabricated on a flexible polyimide substrate for the ex vivo simulation test and in vivo evaluation (as illustrated in Figure 1B–D). The FDMA included a total of six electrodes, which were divided into two groups: group 1 (WE1, CE1, and RE1) and group 2 (WE2, CE2, and RE2). Each group employed a standard three-electrode configuration with a functionalized sensing electrode as the working electrode: a Ag/AgCl electrode as reference electrode and a Pt electrode as counter electrode. The counter electrodes, CE1 and CE2, were obtained by immersing the stainless-steel electrode in a 10 mM H_2_PtCl_6_/0.1 M HCl solution and applying a voltage of −0.4 V for 10 min. The reference electrodes, RE1 and RE2, were made by oxidizing the silver electrode at 0.6 V in 1 M HCl solution. All the prepared electrodes were fixed on the flexible polyimide substrate with conductive paste and cut into lengths of 7 mm ready for use.

## 3. Results and Discussion

### 3.1. Morphology and Structure of Electrode

As shown in Figure 1E, two working electrodes, each with a differential structure, were designed in this study. There was tyrosinase in the PANI/enzyme layer of WE1 and no enzyme in WE2. EDS elemental composition analysis of the two electrodes (Appendix A) showed that the content of Cu in WE1 is 20 times than that in WE2. Because Cu was only contained in tyrosinase, this indicated that tyrosinase was successfully immobilized in WE1.

The surface morphology of the working electrode during preparation was examined with SEM and is shown in Figure 2B–G. Figure 2B shows the SEM image of the SS electrode modified with Au nanostructures, and the enlarged morphology is illustrated in Figure 2C. As can be seen, the Au nanodendrites uniformly covered the electrode surface with a size of about 200 nm. The dendrite-like nanostructure could not only increase the specific surface area of the electrode, but also accelerate the oxidation rate of L-Dopa and improve the detection sensitivity. Figure 2D is the Nafion membrane covered on the Au nanodendrites. A layer of PANI was electropolymerized on the Nafion layer (Figure 2E). The three-dimensional porous structure increased the surface area, and it was very suitable for increasing the amount of enzyme immobilized on the electrode surface [40]. After being modified with tyrosinase and BSA, the electrode was crosslinked in glutaraldehyde vapor, and the morphology after crosslinking is shown in Figure 2F. A layer of thin film on the electrode surface, which was formed during the reaction between glutaraldehyde and the amino or peptide of protein. The covalent bond can provide a chemically robust enzymatic structure to achieve long-term stability for continuous use [31]. In this study, a Nafion membrane was used to separate the enzymatic immobilization layer and the Au nanodendrites. Therefore, L-Dopa could be depleted by tyrosinase in the PANI/enzyme layer of WE1 rather than diffused into the Au nanodendrite catalytic layer and contributed to the electrochemical oxidation current. The electrode was finally coated with PU film. As shown in Figure 2G, the electrode tip is flat and smooth with the PU layer. The good biocompatibility of the PU film can improve the stability of the sensor; meanwhile, the high strength and elasticity of the PU film can protect the electrode structure when it pierces the skin.

**Figure 2 biosensors-12-00102-f002:**
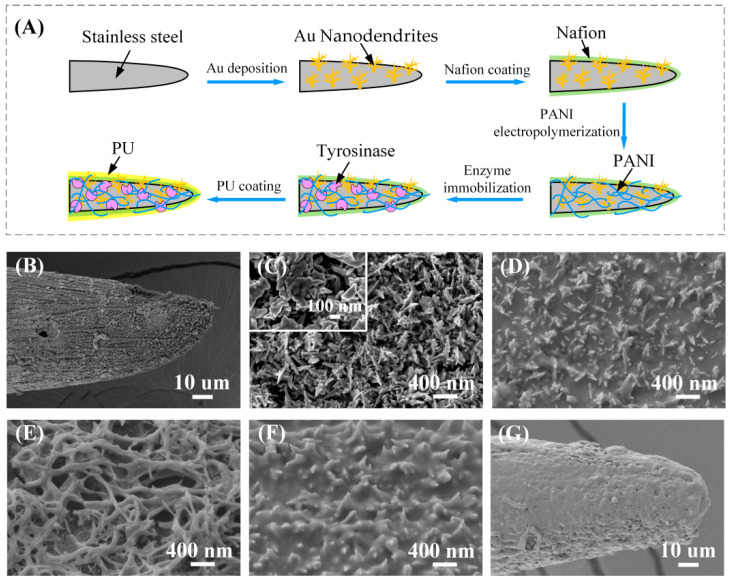
(**A**) The working electrode modification process. (**B**–**G**) The surface morphology of the working electrode during modification. (**B**) Working electrode modified with Au nanostructures. (**C**) Enlarged view of Au nanostructures. (**D**) Nafion membrane covered on the Au nanodendrites. (**E**) PANI nanofibers covered on the Nafion layer. (**F**) The electrode surface morphology after glutaraldehyde vapor crosslinking. (**G**) The electrode morphology after coating with PU film.

### 3.2. Performance of L-Dopa Microneedle Biosensor

Due to the difference in the modified PANI/enzyme layer, the electrochemical responses of the two working electrodes to L-Dopa were different. Figure 3C,D shows the CV response of WE1 and WE2 to L-Dopa in 0.01 M PBS solution (pH = 7.2). It is obvious that WE2 without enzyme modification has a significant oxidation peak to 100 μM L-Dopa, around 0.3 V, while WE1 with tyrosinase modification has no oxidation peak. This is because L-Dopa diffused to WE1 is catalyzed by tyrosinase on the outer layer of the electrode and oxidized to dopaquinone. Therefore, it cannot reach the Au nanodendrite catalytic layer to form an oxidation current (as illustrated in Figure 3A). On the contrary, without the catalysis of tyrosinase, L-Dopa can reach the catalytic layer of WE2 (as displayed in Figure 3B) and form an oxidation peak. AN amount of 0.3 V was chosen as the working potential in this study.

The thickness of the PANI/enzyme layer will affect the diffusion rate of L-Dopa. When the PANI/enzyme layer is too thick, it will block the diffusion of L-Dopa, and the sensitivity of the sensor will decrease. Because the thickness of PANI is related to the electropolymerization time, we studied the relationship between PANI electropolymerization time and sensor response (Appendix A), and selected an optimized electropolymerization time of 100 s.

The amperometric response of WE1 and WE2 to the dynamic concentration change of L-Dopa (0–20 μM) was tested in 0.01 M PBS under the applied potential of 0.3 V; 2 μM L-Dopa was added in PBS every 60 s and the electrode response was recorded. As illustrated in Figure 3E, the current of WE1 remained unchanged, while L-Dopa increased, which validated the idea that WE1 does not respond to L-Dopa. On the contrary, WE2 responded quickly to changes in the concentration of L-Dopa (less than 10 s, as shown in the inset of Figure 3E). We calculated the difference in the current response of WE1 and WE2 and obtained the total response of the differential structure to L-Dopa (as displayed in Figure 3F). The sensitivity of the sensor is 12.6 nA μM^−1^ cm^−2^ (R^2^ = 0.995, *n* = 3), and the limit of detection is 0.18 μM (S/N = 3).

The long-term stability of the sensor was evaluated by storing the two electrodes in 0.01 M PBS (pH = 7.2) at 4 °C for two weeks. The amperometric responses of WE1 and WE2 were tested every few days during storage. According to our previous analysis, WE1 did not respond to L-Dopa due to the modified tyrosinase layer. During the two weeks of the stability test, WE1 did not show any obvious current response to L-Dopa in the concentration of 0–20 μM, indicating that the PU outer layer can maintain enzyme activity for at least two weeks. The sensitivity fluctuation of WE2 (modified with blank enzyme) to L-Dopa is demonstrated in Appendix A. In the first few days, the sensitivity of WE2 showed a slight increase (increasement of 3.3%), and then stabilized. This may be caused by the increased permeability of the outermost biocompatible PU membrane when WE2 was immersed in PBS [41]. After 5 days of soaking, the pore size of PU had stabilized, so the sensitivity of WE2 had also stabilized.

### 3.3. Characterization of Anti-Interference Ability

There are various biomolecules coexisting in human tissue fluid, such as uric acid (UA), ascorbic acid (AA), and glucose. The concentration of these biomolecules in the body is much higher than that of L-Dopa, and they are easily oxidized on the electrode surface and interfere with the sensor signal in the presence of external potential. Therefore, the anti-interference performance of the sensor will affect the accuracy of the detection results. An anti-interference test was conducted in PBS by detecting L-Dopa in the presence of high concentrations of interferents at the applied potential of 0.3 V. The concentration of the biomolecules was as follows: L-Dopa (10 μM), UA (50 μM), AA (50 μM), glucose (200 μM). Figure 4A–D shows the current response of WE1 and WE2 to the mixed solution of L-Dopa and different interferents. We can see that the current response of WE1 is much lower than that of WE2. According to the previous analysis, we know that due to the difference in the PANI/enzyme layer, WE1 could only detect the electrochemical signal of the interferents and WE2 could detect the mixed signal of the interferents and L-Dopa. Therefore, we could extract the current signal related to L-Dopa by subtracting the signals of WE1 and WE2.

Figure 4E displays the histogram of the current response of WE1 and WE2 to different interference mixture solutions and the corresponding current differential values. The sensor’s responses to 10 μM L-Dopa (presented as the current difference between WE1 and WE2) under the coexistence of interference are as follows: 4.98 nA in L-Dopa + UA solution, 4.63 nA in L-Dopa + AA solution, 5.08 nA in L-Dopa + glucose solution, and 4.65 nA in the mixture of L-Dopa + UA + AA + glucose solution. The average response of the sensor is 4.835 nA, and the standard deviation is 4.7%. According to our previous study, in the absence of interference, the current response of the sensor to 10 μM L-Dopa is 4.69 nA (calculated from Figure 3F). The response difference of the sensor in the interference and non-interference solution is less than 5%, indicating that the differential structure provides the sensor with excellent anti-interference ability.

### 3.4. In Vitro Evaluation

The amperometric response of the two working electrodes, WE1 and WE2, was also measured in bovine serum under the applied potential of 0.3 V (Figure 5A). As illustrated in Figure 5B, similar to the response in PBS, the current response of WE2 increased with the increase of L-Dopa concentration and showed a good correlation, while the current of WE1 hardly changed. We noticed that the current baseline of these two electrodes in serum was significantly higher than that of PBS (6 nA in PBS and 27 nA in serum). This is because a large number of biomolecules in the serum may also be oxidized on the surface of the electrode, thereby increasing the oxidation current of the electrode. Meanwhile, the adsorption of biomolecules and proteins on the electrode surface in the serum reduced the diffusion rate of L-Dopa, so the electrode response in the serum took longer to stabilize (about 15 s) than in the PBS (less than 10 s). Although the interferents in serum have an impact on the baseline and response time of the sensor, the sensitivity of the sensor has not changed (as shown in Figure 5C), indicating that whether in PBS or serum, the differential structure can effectively eliminate the influence of interfering substances.

The FDMA was fabricated by assembling two working electrodes with reference and counter electrodes (labeled as group 1: WE1, RE1, CE1 and group 2: WE2, RE2, CE2) on a flexible polyimide base (shows in Figure 1D) for in vivo testing. Before implantation, the performance of the FDMA was assessed by chronoamperometry in bovine serum with a skin model. As shown in Figure 5D, the serum surface was covered with a layer of rat skin, FDMA pierced the skin, and the tip (about 3 mm) was immersed in the serum for the L-Dopa test. The chronoamperometry response of the two groups was recorded by a dual-channel electrochemical workstation at 0.3 V, and the stabilized current signal of both groups is illustrated in Figure 5E. The response of group1 remained unchanged, while the current of group 2 increased linearly with the increase of L-Dopa concentration. The differential current between group 1 and group 2 is proportional to the L-Dopa concentration (Figure 5F). The corresponding calibration curve indicates good linearity, with a correlation coefficient of 0.998 (RSD = 3.1%, *n* = 3). The sensitivity of the FDMA is calculated to be 13 nA μM^−1^ cm^−2^, which is almost the same as the result of the previous separate tests on WE1 and WE2 (12.6 nA μM^−1^ cm^−2^), indicating that the assembled sensor has excellent performance. In this study, the stainless steel is strong enough to pierce through the rat skin, and the PU outer layer can effectively protect the internal structure during penetration. Overall, based on the favorable responses of the in vitro skin model test, the FDMA can be implanted subcutaneously for continuous monitoring of L-Dopa.

### 3.5. In Vivo Evaluation of the Flexible Differential Microneedle Array

After the successful evaluation of the FDMA by chronoamperometry with skin model, we tested the sensor in vivo through minimally invasive monitoring of L-Dopa concentration. As demonstrated in Figure 6A, each group of the FDMA was connected to an electrode channel of the dual electrochemical workstation, respectively, and the current response of both groups was recorded simultaneously. During the measurements, the FDMA was stabilized subcutaneously for approximately 2500 s at a voltage of 0.3 V, and then 4 mL of 8 mM L-Dopa was injected into the rat’s abdominal cavity. The dosage of L-Dopa for the rat was based on previous studies, which was equivalent to 300 mg of L-Dopa for adults [42].

Figure 6B shows the current response of the two electrode groups on FDMA after L-Dopa injection. We noticed that, at the moment of injection, the current response of the two channels showed obvious fluctuation, which might be caused by the stimulation of needle penetration. Due to the abundant blood vessels in the abdominal cavity, L-Dopa was absorbed into the blood vessels soon after injection (approximately 200 s from the response curve), and the concentration of L-Dopa in the subcutaneous interstitial fluid increased gradually. Therefore, by electro-oxidizing L-Dopa to dopaquinone, the current generated from the working electrode of group 2 starts to rise slowly. After approximately 400 s, the response current of group 2 reached the peak value and was maintained for 900 s, and then began to decrease gradually. The electrode of group 1 did not respond to the dynamic change of L-Dopa concentration in the in vivo experiment, which was consistent with the ex vivo skin model experimental results. During the test, the current of the two channels fluctuated by approximately 3800 s and 4200 s. This signal interference was generated by the movement of the rat as the anesthesia gradually metabolized during the test. Although the movement caused fluctuations in the sensor’s response, the amplitude of the interference was almost the same due to the close distance of the two groups, so the interference signal could be eliminated via differential signal processing. Figure 6C shows the response signal after differential correction. By subtracting the signals from the two groups, we could get an interference-free current signal related to the L-Dopa concentration. The in vivo experiment demonstrated the feasibilities of the FDMA for minimal invasive and continuous monitoring of L-Dopa’s dynamic concentration.

## 4. Conclusions

In this study, we proposed a minimal-invasive flexible microneedle array with differential structure for continuous monitoring of L-Dopa for Parkinson’s disease. Compared with previous research, this novel L-Dopa biosensor adapted an opposite strategy where the sensor indirectly detected the signal of L-Dopa by measuring the differential current between two working electrodes, one modified with tyrosinase and the other without it. The differential structure provided the sensor with excellent anti-interference performance and improved the sensor’s accuracy. The fabricated FDMA exhibited favorable analytical performance in serum test and in vitro skin model evaluation with excellent selectivity and a wide linear dynamic range, as well as considerable long-term stability. Ultimately, the flexible microneedle array displayed fast response to the in vivo L-Dopa concentration fluctuations with good anti-interference ability, including the interference from biological molecules and movement.

In general, our research provided an effective method towards drug monitoring and health management for Parkinson’s patients. In future research, we will try to share the reference and counter electrode of the two groups to reduce the number of electrodes and develop a miniaturized microneedle array. Future work will also focus on the further optimization of FDMA, such as the use of flexible electrode materials instead of stainless-steel to improve the wearing comfort of the sensor, and research on biocompatible materials to improve the in vivo stability of the sensor. Future efforts will also concentrate on system integration, developing a miniaturized wireless detection system that integrates with FDMA to realize wearable continuous detection of L-Dopa.

## Figures and Tables

**Figure 1 biosensors-12-00102-f001:**
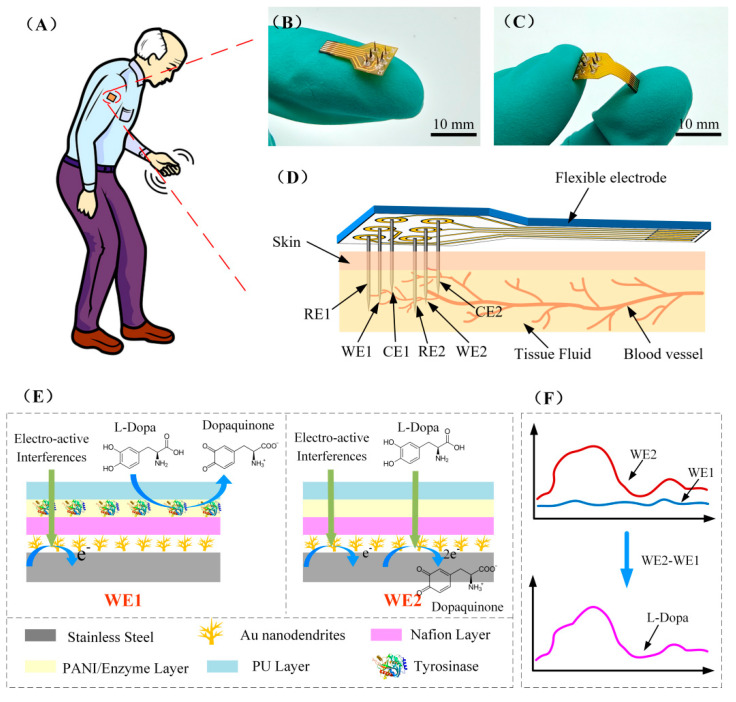
(**A**) Schematic diagram of the minimal-invasive L-Dopa sensor used for the dynamic monitoring of L-Dopa in Parkinson’s patient. (**B**,**C**) Photograph of FDMA. (**D**) Schematic diagram of minimally invasive subcutaneous implantation of FDMA. (**E**) Schematic diagram of the surface modification structure of the working electrode WE1 and WE2, and the reaction process of L-Dopa and other electro-active interferences on the electrode surface. (**F**) The current response signal of WE1 and WE2 and the L-Dopa differential detection principle.

**Figure 3 biosensors-12-00102-f003:**
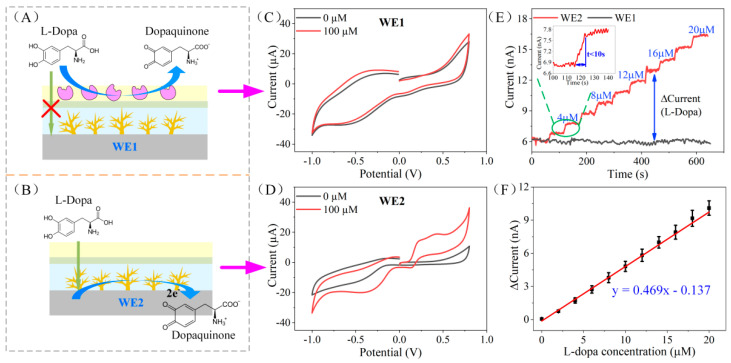
(**A**,**B**) The L-Dopa detection principle of the working electrode WE1 and WE2. (**C**,**D**) The CV response of WE1 and WE2 to 0 μM and 100 μM L-Dopa in 0.01 M PBS solution. (**E**) The amperometric response of WE1 and WE2 to the dynamic concentration change of L-Dopa (0–20 μM). The inset is an enlarged view of the amperometric response of WE2 from 100 s to 140 s. (**F**) Calibration curve of the L-Dopa concentration vs. current response difference of WE1 and WE2.

**Figure 4 biosensors-12-00102-f004:**
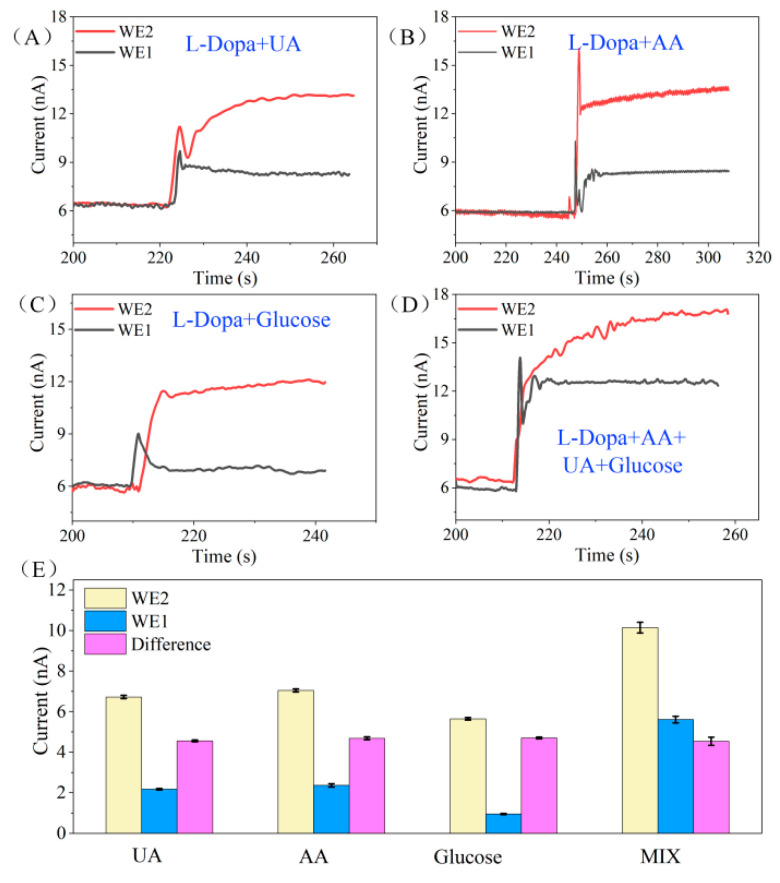
(**A**–**D**) The current response of WE1 and WE2 to the mixed solution of L-Dopa and different interferents. The concentration of the biomolecules are as follows: L-Dopa (10 μM), UA (50 μM), AA (50 μM), glucose (200 μM). (**E**) The histogram of current response of WE1 and WE2 to different interference mixture solutions and the corresponding current differential values. UA: 10 μM L-Dopa + 50 μM UA, AA: 10 μM L-Dopa + 50 μM AA, Glucose: 10 μM L-Dopa + 200 μM Glucose, MIX: 10 μM L-Dopa + 50 μM UA + 50 μM AA + 200 μM Glucose.

**Figure 5 biosensors-12-00102-f005:**
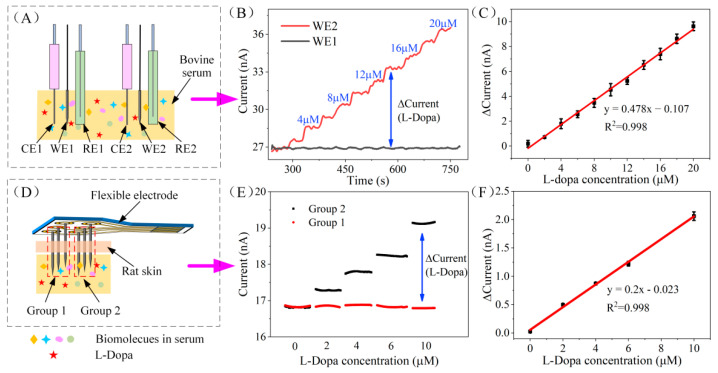
(**A**) Schematic diagram of in vitro serum evaluation of WE1 and WE2. (**B**) The amperometric response of WE1 and WE2 to the dynamic concentration of L-Dopa (0–20 μM) in bovine serum. (**C**) Calibration curve of the L-Dopa concentration vs. current response difference of WE1 and WE2 in bovine serum. (**D**) Schematic diagram of in vitro skin model evaluation of the assembled FDMA. (**E**) Chronoamperometry response of the FDMA recorded by a dual channel electrochemical workstation at 0.3 V. (**F**) Calibration curve of the L-Dopa concentration vs. current response difference of the FDMA in the skin model test.

**Figure 6 biosensors-12-00102-f006:**
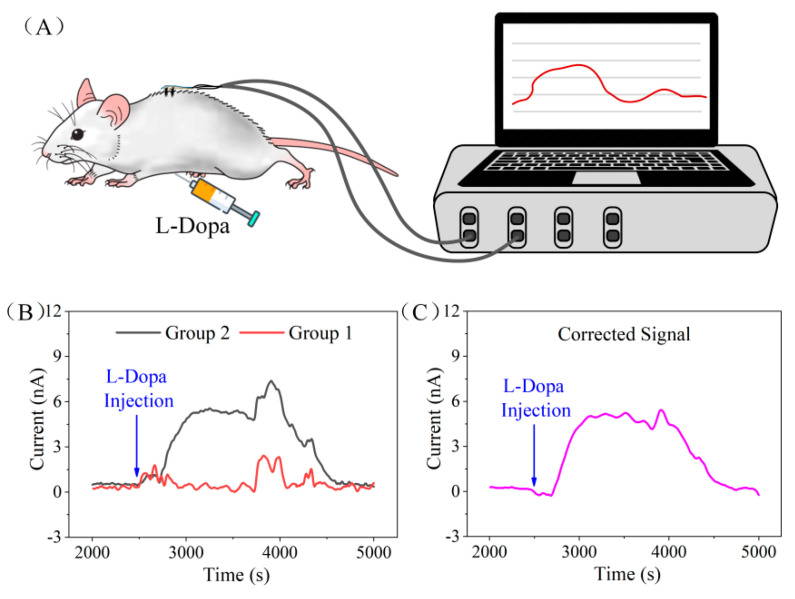
(**A**) The in vivo test schematic diagram of the minimal−invasive flexible microneedle array. (**B**) Current response of the two electrode groups on flexible microneedle array after L-Dopa injection. (**C**) The flexible microneedle array current response signal to L-Dopa after differential correction.

## Data Availability

Not applicable.

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
