# Peer review of "Differential Amperometric Microneedle Biosensor for Wearable Levodopa Monitoring of Parkinson’s Disease"

_biosensors, 2022, doi:10.3390/bios12020102_

Round 1

Reviewer 1 Report

The interference of electroactive species in physiological environment is one of the biggest challenges for electrochemical detection in vivo, especially when the concentration of the tested substance is very low. In this manuscript, the authors proposed a novel minimal-invasive L-Dopa biosensor based on flexible differential microneedle array (FDMA). The differential structure provided the sensor with good anti-interference ability and improved the sensor’s accuracy. The FDMA L-Dopa sensor also displayed a fast response to in-vivo L-Dopa dynamic concentration in animal experiment. Overall, the manuscript is well documented and I recommend this paper to be accepted after a minor revision. 1. The role of gold nanoparticles on the electrode surface should be explained? 2. The author claimed that “This differential approach effectively addressed the challenges for low-concentration substances detection under the influence of high-concentration interfering substances”, so what is the concentration range of L-Dopa and interfering substances in vivo? 3. The unit in Figure 3 C and D, “um” should be “μM”. 4. Page 6, line 221, “The thickness of the enzyme layer will affect the diffusion rate…”, I think it should be “The thickness of the PANI layer…” 5. About the dendrite-like gold nanoparticles, the author use “Au nanoflowers” in figure 1, however they use “Au nanodendrites” in figure 2. They should use one description. 6. The tenses in the manuscript should be consistent, and there are cases where multiple tenses are mixed in some places.

Author Response

Thank you so much for your time evaluating our manuscript and we gratefully respect these comments and attentively took these comments into consideration. Here are our responses to the comments.

Comment 1. The role of gold nanoparticles on the electrode surface should be explained?

Response: Gold nanoparticles are nanostructures with catalytic ability. When the electrochemically active substances in the body (such as L-Dopa, ascorbic acid and uric acid) reach the surface of the gold nanoparticles, they will be catalyzed by the gold nanostructures under the applied voltage and generate an oxidation current.

Comment 2. The author claimed that “This differential approach effectively addressed the challenges for low-concentration substances detection under the influence of high-concentration interfering substances”, so what is the concentration range of L-Dopa and interfering substances in vivo?

Response: The concentration range of L-Dopa is about 0-15μM for Parkinson’s Disease patients according to this paper. (Eneurologicalsci, 2018, 13: 8-13.)

Normal blood glucose level for non-diabetics is 3.5-5.5 mM. (Archives of disease in childhood, 2016, 101(6): 569-574.)

The ascorbic acid concentration range is 0.50–17.40 mg/L (2.8-98μM).(Journal of International Medical Research, 2018, 46(1): 168-174.)

The uric acid concentration range is 120-380μM. (Journal of Pharmacology and Experimental Therapeutics, 2008, 324(1): 1-7.)

So, compared with interfering substances, the L-Dopa concentration is much lower.

Comment 3. The unit in Figure 3 C and D, “um” should be “μM”.

Response: Thank you so much for your comment, we have revised the units in Figure 3 C and D.

Comment 4.  Page 6, line 221, “The thickness of the enzyme layer will affect the diffusion rate…”, I think it should be “The thickness of the PANI layer…”

Response: In our study, tyrosinase was immobilized in the PANI layer, so we mixed used the enzyme layer and the PANI layer, we are very sorry about that. In the revised manuscript, we changed enzyme layer and PANI layer into PANI/enzyme layer.

Comment 5. About the dendrite-like gold nanoparticles, the author use “Au nanoflowers” in figure 1, however they use “Au nanodendrites” in figure 2. They should use one description.

Response: We are very sorry for the inconsistency in the term for gold nanostructures, we have corrected this mistake in the revised version and used “Au nanodendrites” to present the gold nanostructures.

Comment 6. The tenses in the manuscript should be consistent, and there are cases where multiple tenses are mixed in some places. 

Response: Thank you very much for your comments, we have revised the tenses.

Reviewer 2 Report

This manuscript proposed a differential amperometric microneedle biosensor for wearable L-Dopa monitoring of Parkinson’s disease. The differential structure provided the sensor with good anti-interference ability and improved the sensor’s accuracy, making a promising application of minimal-invasive L-dopa monitoring, which is very meaningful for Parkinson’s patients to control drug dosage and reduce their complications. The proposed flexible microneedle biosensor exhibits a good analytical performance with wide linear dynamic range, high sensitivity, long-term stability as well as fast response to L-Dopa dynamic concentration fluctuation in-vivo. I suggest to accept the manuscript after a minor revision.

  1. The author should add a scale bar in figure1 B and C.
  2. In section 2.1, the diameter of aluminum oxide should be provided.
  3. Does the immobilization of enzyme affect the diffusion of substrates?
  4. There are two groups of electrode array with respective reference electrode and counter electrode. Does it possible to share one reference electrode of the two group?
  5. The English writing should be improved, there are cases where multiple tenses are mixed in some places. “Au nanodendrites” and “Au nanoflowers” were used to describe the dendrite-like gold nanoparticle. The author should use a unified description.
  6. The author also missed some wearable-based sweat sensors such as Nano Lett. 2021, 21 (20), 8880-8887; ACS Sens 2020, 5 (6), 1548-1554; Biosensors and Bioelectronics 2022, 196, 113760.

Author Response

Thank you so much for your time evaluating our manuscript and we gratefully respect these comments and attentively took these comments into consideration. Here are our responses to the comments.

Comment 1. The author should add a scale bar in figure1 B and C.

Response: Thank you very much for your comments, we have added scale bar in Figure B and C.

Comment 2. In section 2.1, the diameter of aluminum oxide should be provided

Response: We have added the diameter of aluminum oxide in section 2.1.

Comment 3. Does the immobilization of enzyme affect the diffusion of substrates?

Response: Yes, the immobilization of enzyme would affect the diffusion of substrates. When the PANI/enzyme layer is too thick, L-Dopa could not pass through the PANI/enzyme layer and the sensitivity of the sensor will decrease. We studied the relationship between the thickness of PANI/enzyme layer and the sensor’s sensitivity and chosen the optimized thickness. It was demonstrated in section 3.2 and marked with red.

Comment 4. There are two groups of electrode array with respective reference electrode and counter electrode. Does it possible to share one reference electrode of the two group?

Response: Yes, it is possible to share one reference electrode of the two groups because the two groups worked at the same potential. However, in the current experimental stage, in order to avoid mutual interference between the two groups, we chose a separate reference electrode for them. In future research, we will try to share the reference electrode of the two groups to reduce the number of electrodes and develop a miniaturized microneedle array.

Comment 5. The English writing should be improved, there are cases where multiple tenses are mixed in some places. “Au nanodendrites” and “Au nanoflowers” were used to describe the dendrite-like gold nanoparticle. The author should use a unified description.

Response: Thank you very much for your comments, we have revised the tenses.

We are very sorry for the inconsistency in the term for gold nanostructures, we have corrected this mistake in the revised version and used “Au nanodendrites” to present the gold nanostructures.

Comment 6. The author also missed some wearable-based sweat sensors such as Nano Lett. 2021, 21(20), 8880-8887; ACS Sens 2020, 5 (6), 1548-1554; Biosensors and Bioelectronics 2022,196, 113760.

Response: We have added the mentioned references.

Reviewer 3 Report

In this manuscript, the authors proposed a novel minimal-invasive L-Dopa biosensor based on flexible differential microneedle array (FDMA). The differential structure provided the sensor with good anti-interference ability. The FDMA L-Dopa sensor also displayed a fast response to in-vivo L-Dopa dynamic concentration in animal experiment. Overall, the method proposed in this paper is acceptable. My suggestion regarding this manuscript are as follows:

  1. Electrons are also generated when tyrosinase catalyzes the conversion of levodopa to dopaquinone. Why don’t these electrons affect the electrode signal?
  2. The diameter of the microneedle should be provided in the Materials and Methods section.
  3. The abbreviation “MIX” in Figure 4E should be given an explanation in the text or figure caption.
  4. Some English writing should be improved. For example, the tenses in the article should be consistent.

Author Response

Thank you so much for your time evaluating our manuscript and we gratefully respect these comments and attentively took these comments into consideration. Here are our responses to the comments.

Comment 1. Electrons are also generated when tyrosinase catalyzes the conversion of levodopa to dopaquinone. Why don’t these electrons affect the electrode signal?

Response: Thank you for your comments.The tyrosinase was immobilized in the PANI/enzyme layer, and the inner layer of the PANI/enzyme layer was the Nafion layer which was used to separate the enzymatic immobilization layer and the Au nanodendrites. The L-Dopa depleted in the enzyme layer rather than diffused into the electrode surface and contributing to the electrochemical oxidation current like the other electrode without tyrosinase immobilization. The electrons and hydrogen generated by tyrosinase catalyzing L-dopa were combined with oxygen to generate H2O2. Therefore, the enzymatic catalyzing process of L-dopa do not affect the signal at the working electrode.

Comment 2. The diameter of the microneedle should be provided in the Materials and Methods section.

Response: Thank you for your comments, we have added the diameter of the microneedle in the Materials and Methods section.

Comment 3. The abbreviation “MIX” in Figure 4E should be given an explanation in the text or figure caption.

Response: We have added the explanation in the legend of Figure 4E and marked it with red.

Comment 4. Some English writing should be improved. For example, the tenses in the article should be consistent.

Response: Thank you so much for your comments, we have revised the tenses and our English writing.

Reviewer 4 Report

The present manuscript reported “Differential Amperometric Microneedle Biosensor for Wearable Levodopa Monitoring of Parkinson’s Disease”. The paper is interesting in bio-sensing investigation, microfabrication application. Clearly, enough tests plus in-vitro skin model and in-vivo evaluation have been done, but it needs more medications

 Therefore, I can recommend it for publication after improvement with minor revision!

  1. Differential strategy is a good way to decrease the potential interferences, please explain it more, Also, could you predict the potential interface.
  2. page5, it is not clear why you used Nafion, you mentioned, “Nafion layer is used to isolate the electrons generated during the oxidation of L- Dopa by tyrosinase.” But I think it is not completely true, or why Nafion can isolate electrons, but, PANI or Gold can not do it! why did you choose them, please explain all electrode components clearly and separately.
  3. please add time delay (response time) after injection DA till recording signal (in vivo experiment)
  4. please add, the regression coefficient for both curves, Figure 5. C. and F.
  5. page 6, about stability, As I found, you tested it in PBS, it is acceptable. But please add more information about the stability of electrodes during in vivo measurement.
  6. please add these references to improve your work

--- Bian, Sumin, Bowen Zhu, Guoguang Rong, and Mohamad Sawan. "Towards wearable and implantable continuous drug monitoring: A review." Journal of Pharmaceutical Analysis 11, no. 1 (2021): 1-14.

---Erdem, Özgecan, Ismail Eş, Garbis Atam Akceoglu, Yeşeren Saylan, and Fatih Inci. "Recent Advances in Microneedle-Based Sensors for Sampling, Diagnosis and Monitoring of Chronic Diseases." Biosensors 11, no. 9 (2021): 296.

--Kumar, Saurabh, Chandra Mouli Pandey, Amir Hatamie, Abdolreza Simchi, Magnus Willander, and Bansi D. Malhotra. "Nanomaterial‐Modified Conducting Paper: Fabrication, Properties, and Emerging Biomedical Applications." Global Challenges 3, no. 12 (2019): 1900041.

--Oung, Qi Wei, M. Hariharan, Hoi Leong Lee, Shafriza Nisha Basah, Mohamed Sarillee, and Chia Hau Lee. "Wearable multimodal sensors for evaluation of patients with Parkinson's disease." In 2015 IEEE International Conference on Control System, Computing and Engineering (ICCSCE), pp. 269-274. IEEE, 2015.

Author Response

Thank you so much for your time evaluating our manuscript and we gratefully respect these comments and attentively took these comments into consideration. Here are our responses to the comments.

Comment 1. Differential strategy is a good way to decrease the potential interferences, please explain it more, Also, could you predict the potential interface.

Response: Thank you for your comments.

The differential strategy in our design utilizing two working electrodes. One working electrode is response to L-Dopa and interferences, by directly electrochemical oxidizing L-Dopa and these electroactive interferences, while the other working electrode is only response to these electroactive interferences based on enzymatic depletion of L-Dopa by tyrosinase. The differential current response of these two electrodes is related to the concentration of L-Dopa with eliminating the common mode interference. We also improved the explanation in our revised manuscript.

The potential interface includes these electroactive substrates, such as uric acid, ascorbic acid, glucose, acetaminophen and other electroactive medicines. In our current study, we take the anti-interference experiment with uric acid, ascorbic acid, glucose. We will test more interferences in our future study.

Comment 2. page5, it is not clear why you used Nafion, you mentioned, “Nafion layer is used to isolate the electrons generated during the oxidation of L- Dopa by tyrosinase.” But I think it is not completely true, or why Nafion can isolate electrons, but, PANI or Gold can not do it! why did you choose them, please explain all electrode components clearly and separately.

Response:Thank you for your comments and we apologize for our inappropriate description. In fact, the Nafion layer is used to separate the enzymatic immobilization layer and the Au nanodendrites, making the L-Dopa depleted in the enzyme layer rather than diffused into the electrode surface and contributing to the electrochemical oxidation current like the other electrode without tyrosinase immobilization. 

The layer structure of the two working electrodes, WE1 and WE2, are Au nanodendrite catalytic layer, Nafion layer, PANI/enzyme layer, and PU layer. The difference between WE1 and WE2 is the PANI/enzyme layer. In WE1, the tyrosinase was immobilized in the PANI/enzyme layer. The tyrosinase is used to catalytic oxidation depilate L-Dopa. While in WE2, we use BSA to replace tyrosinase, so L-Dopa is not depilated and is diffused into the electrode surface, generating the electrochemical oxidation current response.

We choose Nafion as the separation layer also because that Nafion is widely used as a selectively permeable membrane, which can decrease the current response of interference like uric acid and ascorbic acid.

Thank you again for your kind comment. We have corrected our description and added the explanation of the electrode structure in the revised manuscript.

Comment 3. please add time delay (response time) after injection DA till recording signal (in vivo experiment)

Response: We have added the time delay in our revised manuscript, the time delay after injection DA till recording signal is 200 s.

Comment 4. please add, the regression coefficient for both curves, Figure 5. C. and F.

Response: The regression coefficients were added in Figure 5. C. and F.

Comment 5. page 6, about stability, As I found, you tested it in PBS, it is acceptable. But please add more information about the stability of electrodes during in vivo measurement.

Response: Thanks for your understanding and comment. The in vivo stability is very important for implantable biosensor. However, it is also an extremely challenging technology. In our current in-vivo experiment, the total experiment time was 2 hours, after that the microelectrodes were taken off from the rat because it is difficult to stably fixed on the back of the rat then it was awake. The microelectrodes were prone to failure due to frictional movement and protein adsorption. We will improve the mechanical strength of the electrode and resistance to protein adsorption of the electrode surface, and carry out the in-vivo biocompatibility and stability experiment.

Comment 6. please add these references to improve your work

Response:Thank you for your comments. These important references have been cited in the revised manuscript.